# Tungsten-Bearing Wodginite from the Kester Deposit, Eastern Siberia, Russia

**Viktor I. Alekseev \*** and **Ivan V. Alekseev**

Faculty of Geological Prospecting, Saint-Petersburg Mining University, 2, 21st Line, St. Petersburg 199106, Russia
* Correspondence: alekseev_vi@pers.spmi.ru

**Abstract:** Li-F granites from the Kester deposit (Yana Plateau in Yakutia, Russia) are proved to be connected with a rare-metal complex of accessory minerals: montebrasite, columbite-(Mn), columbite-(Fe), tantalite-(Mn), Ta-bearing cassiterite, U-bearing microlite, W-bearing ixiolite, niobian ferberite, U–Hf-rich zircon, and Ta-bearing rutile. Accessory wodginite was discovered at depths of up to 150 m in association with tantalite-(Mn), columbite-(Mn), and cassiterite. According to the content of $WO_3$ (1.23%–3.33%) and the values of $Mn/(Mn + Fe_t)$ and $Ta/(Ta + Nb)$, Yakut wodginite is an intermediate mineral between wodginite and a hypothetical mineral of the wodginite group—"wolframowodginite". The discovery of tungsten-bearing wodginite at the Kester deposit confirms the widespread presence of tungstic and tungsten-bearing accessory minerals in Li-F granites in the Russian Far East. It also serves as an indicator of rare-metal tin-tantalum-bearing granites and pegmatites.

**Keywords:** wodginite group; wolframowodginite; lithium-fluoric granite; tungsten; tantalum; Kester deposit; Yakutia; Eastern Siberia; Russia

## 1. Introduction

The development of the metallurgy and battery industry determines the steady growth in the consumption of tantalum. In a report on critical raw materials, the European Commission notes the crucial importance and the shortage of tantalum raw materials, the uneven geographical distribution of their reserves, the limited scope of recycling, and the challenging competition for mining and marketing [1]. One of the industrial minerals of tantalum—wodginite $MnSnTa_2O_8$—is relatively rare [2–4]. There are three important peculiarities in its current study:

1. Minerals of the wodginite group (WGMs) are known as accessories in rare-metal pegmatites. In the past 20 years, there have been new publications on such accessory minerals in rare-metal Li-F granites from Sn–W–Ta deposits in Algeria, Egypt, Spain, China, and the Czech Republic [5–16].
2. As industrial minerals, WGMs are of practical interest; there are more than 79 points of their manifestation [17]. However, in Russia, among hundreds of rare-metal manifestations, we only know of four localization points of WGMs: Kola Peninsula, Eastern Sayan Mountains, the Urals, and Primorye [3,18].
3. In addition to the approved WGMs—titanowodginite, ferrowodginite, etc. [2]—in 1998, a new mineral species called "wolframowodginite" was described in pegmatites in Canada [19]. Later, it was also found in pegmatites in India [20] and in granites in China [11,12].

These circumstances bring us to write a paper that introduces new data on tungsten-bearing wodginite found in lithium-fluoric granites (LFGs) from the Kester deposit (Eastern Siberia, Russia).

## 2. Geological Setting

The Kester deposit is related to the Arga Ynnakh Khaya granite pluton (Figure 1), which is located in the north of Verkhoyansk Fold Belt (Yana Plateau, Eastern Yakutia, Russia). The pluton is embedded within sandstones and siltstones of the Ladinian and the Carnian Stages of the Upper Triassic, which have been crumpled into large folds. Due to erosion, two domes have been exposed on the surface, the eastern and western domes, with areas of 33.7 and 9.7 km$^2$, respectively. The western dome is composed of medium- and coarse-grained andesine granites and granodiorites dated at 137 Ma. The andesine granites of the western dome contain the Kester harpolith [21]. Its crescent outcrop, which is 2300 m long, is composed by LFG–microcline–albite rock ("white granites") made of lithium mica, topaz, and montebrasite. Montebrasite–albite granites prevail near the surface, whereas at depths of over 60 m, we observe the predominance of muscovite–topaz–albite varieties. LFGs are sugary white, coarse-grained rocks with a "snow-ball" quartz texture dated at 129 Ma. The hanging wall of the harpolith is complicated by a marginal zone of druse-like pegmatites, i.e., stocksheiders, that are 0.25–0.5 m thick and pegmatoid schlieres of up to 0.4 m in diameter. In the western near-contact zone of the "white granite" harpolith, a body of topaz–mica–quartz greisen is localized. The greisen body (containing cassiterite; columbite; tantalite; phosphates of Li, Al, Ca, and Cu; and sulfides of Fe, Sn, and Cu) forms the Kester Li–Ta–Sn deposit.

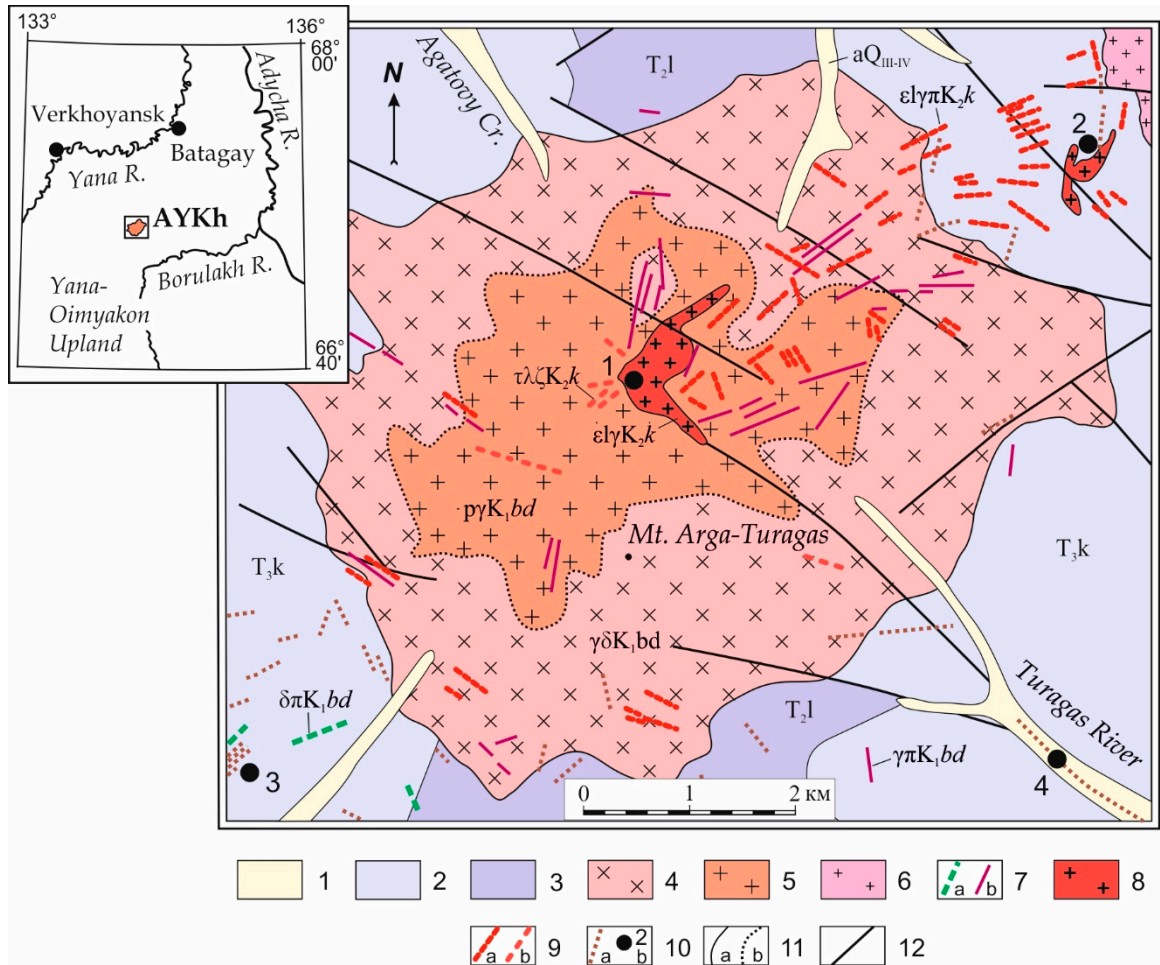

**Figure 1.** Geological map of the Arga Ynnakh Khaya pluton: (1) Alluvial and talus solifluction boulder–pebble sediments, gravels, sand, and loamy sands (aQ$_{III-IV}$). (2) Carnian sandstones and siltstones (T$_3$k). (3) Ladinian siltstones and sandstones (T$_2$l). (4–7) Baky-Derbeke Complex: (4) main

granodiorite ($\gamma\delta K_1 bd$), (5) main andesine granite ($p\gamma K_1 bd$), (6) biotite, two-mica granites, and leucogranites ($\gamma K_1 bd$), (7) dikes of (a) diorite–porphyrite ($\delta\pi K_1 bd$) and (b) granite–porphyry, leucogranite–porphyry, and aplite ($\gamma\pi K_1 bd$). (8,9) Kester Complex: (8) fine- to coarse-grained albite granite ($\varepsilon l\gamma K_2 k$) and (9) dikes of (a) fine-grained albite granite and (b) ongonite ($\tau\lambda\zeta K_2 k$). (10) Mineralized zones (a) and rare-metal and tin–ore occurrences (b): (1) Kester, (2) Tumannoye, (3) Ulakhan-Egelyakh, and (4) Turagas placer. (11) Geological boundaries between (a) rocks of different ages and (b) facies varieties. (12) Faults. Geographical location of the Arga Ynnakh Khaya pluton (AYKh) is shown in the insertion.

Close to the Kester deposit, other points of rare-metal and tin mineralization such as quartz veins, metasomatic zones, and cassiterite placers along the channels of Turagas River, and Ilin-Salaya and Agatovy Creeks are present. Dikes and stock-like intrusions of albite granite, to which the Tumannoe and Ytyr-Khalan occurrences are related, are found at the outer contact (Figure 1) [21,22].

In chemical composition, LFGs are felsic (67.11–72.95 wt% $SiO_2$), subalkaline (7.27–8.84 wt% $K_2O + Na_2O$), and rich in Al (14.61–17.84 wt% $Al_2O_3$). They belong to the high-phosphorous type (1.14–2.70 wt% $P_2O_5$) and are enriched with F (0.8–2.2 wt%), as well as Li, Rb, Cs, Ta, Nb, Sn, W, Zr, REE, and Y. Their geochemistry is reflected in an accessory assemblage. Topaz, montebrasite, Ta-bearing cassiterite, fluorapatite, columbite-(Mn), and tantalite-(Mn) are the major accessories (0.1–0.5 vol%); niobian ferberite, W-bearing ixiolite, U–Hf-rich zircon, Ta-bearing rutile, ilmenite, spodumene, monazite-(Ce), xenotime-(Y), and uraninite are minor ones (<0.1 vol%). Accessory minerals are included in the aggregates of lepidolite, topaz, and montebrasite.

Main Ta–Nb oxides: Columbite group minerals mainly present with tabular crystals of columbite-(Mn) and columbite-(Fe) of lengths of 15 to 150 µ, either in albite aggregates or as inclusions in mica and topaz. Tantalite-(Mn) forms 1–20 µ thick rims, which surround columbite, or xenomorphic microinclusions of less than 8 µm in thickness in topaz and cassiterite.

Most often, the columbite group is represented by polymineral individuals with concentric growth zonality (from the core to the rim): columbite-(Fe) $\rightarrow$ columbite-(Mn) $\rightarrow$ tantalite-(Mn) + U-bearing microlite $\rightarrow$ columbite-(Mn) [23]. The generalized empirical formula of columbite–tantalite of the Kester deposit is $(Mn_{0.58}Fe_{0.37})_{0.95}(Nb_{1.39}Ta_{0.55}Ti_{0.04}W_{0.03}Sn_{0.01})_{2.02}O_6$.

## 3. Materials and Methods

The paper presents the results of a study on the reference collection of granitoids collected during the geological mapping of the Arga Ynnakh Khaya pluton at the scale of 1:50,000 and the exploration of the Kester deposit, a total of 155 samples. The petrographic study (Materials Analysis microscope Leica DM 2500 M, Germany) and the geochemical analysis of unaltered granitic rocks that were performed at Core Resource Center of St. Petersburg Mining State University (X-ray fluorescence and mass spectrometry with inductively coupled plasma (Shimadzu Lab Center XRF-1800 wavelength dispersive X-Ray fluorescence spectrometer; Shimadzu ICPE-9000 multitype ICP emission spectrometer)) revealed LFGs with increased contents of Li, Rb, Cs, Ta, Nb, Sn, W, Zr, REE, and Y. Thin, polished sections of these rocks were studied using optical and electron microscopy to find rare-metal-bearing minerals. Electron microscopy was performed using a JEOL JSM-6510LA scanning electron microscope equipped with a JED-2200 energy dispersive spectrometer at Institute of Precambrian Geology and Geochronology, Russian Academy of Sciences in St. Petersburg. The operation conditions were 20 kV accelerating voltage and 1.5 nA current intensity; ZAF correction was used for matrix effects.

W-bearing ixiolite, tungsten-bearing columbite, niobian ferberite, and micrograins of tungsten-bearing wodginite were found during the detailed study of tantalum and tungsten minerals from LFGs [23]. Quantitative chemical analysis of wodginite (17 samples) were performed on polished carbon-coated sections using JXA-8230 Electron Probe Microanalyzer working in wavelength-dispersion mode at St. Petersburg Mining University. The

system was operated using an accelerating voltage of 20 kV, a beam current of 100 nA, a spot size of 3 μm, and a counting time of 30 s on the peaks and 10 s on the backgrounds. Matrix effects were corrected using the ZAF procedure. The following minerals were used as standards: $LiTaO_3$ (Ta Lα), $LiNbO_3$ (Nb Lα), spessartine (Mn Kα), $SnO_2$ (Sn Lα), fayalite (Fe Kα), $CaWO_3$ (W Lα), TiO (Ti Kα), diopside (Ca Kα), Sc metal (Sc Kα), and zircon (Si Kα, Zr Lα, Hf Lα).

The diagnostics of tungsten-bearing wodginite was performed taking into account the papers [2,19,24]. The relationships between wodginite and other minerals, the anatomy of individuals, and the concentration and distribution of trace elements in crystals and enclosed minerals were noted [25,26].

## 4. Results

Accessory minerals from LFGs from the Kester deposit compositionally correspond to the main industrial types of ore mineralization. The principal paragenesis is formed by tantaloniobates, Nb-Ta-bearing cassiterite, and tungsten-bearing minerals (W-bearing ixiolite, tungsten-bearing columbite, and Nb-Ta-bearing ferberite). Minerals of the wodginite group have some tungsten (on average, 2.67 wt% $WO_3$) [27].

### 4.1. Wodginite in Lithium-Fluoric Granites from the Kester Deposit

In LFGs from the Kester deposit, accessory wodginite was found on the surface outcrops and in the wells at depths of 13–150 m. The mineral presented with anhedral grains ranging in size from 0.7 to 10 μm in aggregate with tantalite-(Mn), which formed rims around columbite-(Mn). Along with wodginite, rims contained U-bearing microlite (Figure 2a–d). Subhedral tabular and xenomorphic grains of sizes of 1–19 μm, building up on tantalite-(Mn), columbite-(Mn), and columbite-(Fe), were also very characteristic (Figure 2e,f). Crystals of accessory cassiterite with Ta and Nb contained inclusions of columbite-(Mn), surrounded by rims of wodginite of 0.7–8 μm in width, along the borders of the host mineral. Wodginite in cassiterite from greisens was only found in a few cases on the border, with cassiterite containing inclusions of tantalite-(Mn). The rims of wodginite in greisens were relatively wide, up to 17 μm.

### 4.2. Composition of Wodginite from the Kester Deposit

Wodginite from the Kester deposit was enriched with manganese (Mn/(Mn + $Fe_t$) = 0.61–0.82). In comparison with typical wodginite, it had less tantalum (Ta/(Ta + Nb) = 0.69–0.84), $TiO_2$, and $SnO_2$ (Table 1; Figure 3). The contents of typomorphic components $Nb_2O_5$ and $WO_3$ in wodginite were, on average, 12.05% and 2.46%, respectively (Figure 4). The generalized empirical formula is $(Mn^{2+}, Fe^{2+})_{1.00}(Sn, Ta, Ti, Fe^{3+})_{1.00}(Ta, Nb, W^{6+})_{2.00}O_8$.

The closest analog of wodginite from Yakutia is wodginite in LFGs from the large tungsten Dajishan deposit in Southeast China [12]. Characteristic features of wodginite from LFGs, such as high contents of $Nb_2O_5$ and $WO_3$, have a stronger manifestation in wodginite from the Songshugang Nb-Ta-W-Sn deposit in China [11] (Table 1). The wodginite under consideration is close in composition to titanowodginite (Figures 3 and 4).

According to the content of $WO_3$, as well as to the values of (Mn/(Mn + $Fe_t$)) and (Ta/(Ta + Nb)), wodginite from Yakutia is an intermediate mineral between wodginite and a hypothetical mineral of the wodginite group—"wolframowodginite" [19] (Figures 3 and 4; Table 1). It should be noted that traces of tungsten also occurred in other tantalo-niobates of the rare-metal granites from the Kester deposit: 0.12–5.23 wt% in columbite-(Mn), 0.90–7.12 wt% in columbite-(Fe), 0.19–4.72 wt% in tantalite-(Mn), and 0.19–1.98 wt% in microlite [23]. Wodginite from greisens that accompanied granites did not contain any traces of tungsten.

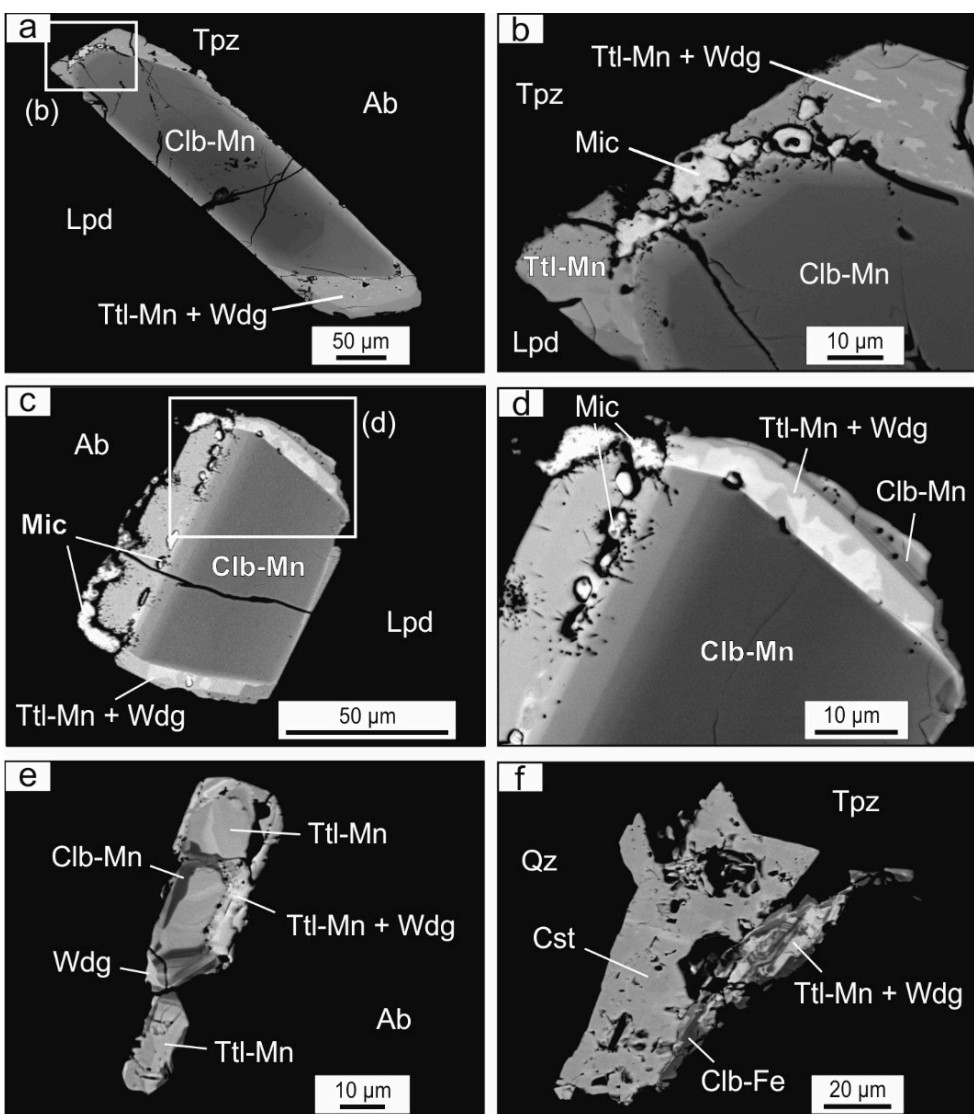

**Figure 2.** BSE images of wodginite from lithium-fluoric granites from the Kester deposit (East Yakutia): (**a**,**c**) Euhedral W-bearing columbite-(Mn) with a rim of aggregated W-bearing tantalite-(Mn), wodginite, and microlite. (**b**,**d**) Details of Figures (**a**,**c**): rim of complex aggregate formed by tantalite-(Mn), wodginite, and microlite in columbite-(Mn). (**e**) Subhedral tantalite-(Mn) with a nucleus of columbite-(Mn) and a rim of a complex aggregate made of tantalite-(Mn) and W-bearing wodginite. (**f**) Partial pseudomorphs of tantalite-(Mn) and wodginite after crystals of tungsten-bearing columbite-(Fe) that form a solid aggregate with Nb-bearing cassiterite. Indices of minerals were taken from Warr (2021) [28].

**Table 1.** Chemical composition (wt%) of tungsten-bearing wodginite in Li-F granites from the Kester deposit and other deposits across the world.

| Component | Wodginite from the Kester Deposit | | | Tungsten-Bearing Wodginite | | | | Wodginite Group | |
|---|---|---|---|---|---|---|---|---|---|
| | **WK-208 *** | **WK-222 *** | **WK-av *** | **1 *** | **2** | **3** | **4 *** | **Wwdg** | **Wdg *** |
| $Li_2O$ | – | – | – | – | – | – | 0.21 | 0.06 | – |
| MnO | 9.63 | 9.81 | 10.06 | 11.54 | 10.88 | 8.69 | 8.33 | 11.57 | 9.85 |
| $FeO_t$ [(1)] | 5.40 | 5.47 | 5.25 | 1.66 | 0.87 | 3.52 | 8.85 | 4.97 | 2.79 |
| $TiO_2$ | 1.01 | 0.20 | 1.37 | 1.05 | 1.45 | 0.92 | 1.68 | 0.70 | 1.71 |

**Table 1.** *Cont.*

| Component | Wodginite from the Kester Deposit | | | Tungsten-Bearing Wodginite | | | | Wodginite Group | |
|---|---|---|---|---|---|---|---|---|---|
| | WK-208 * | WK-222 * | WK-av * | 1 * | 2 | 3 | 4 * | Wwdg | Wdg * |
| $Nb_2O_5$ | 13.80 | 13.40 | 12.05 | 14.08 | 4.92 | 5.08 | 17.98 | 10.39 | 8.11 |
| $SnO_2$ | 11.27 | 11.25 | 11.81 | 9.52 | 17.50 | 12.96 | 4.35 | 9.37 | 13.33 |
| $Ta_2O_5$ | 55.56 | 58.89 | 58.83 | 57.95 | 62.94 | 61.33 | 38.97 | 46.27 | 63.13 |
| $WO_3$ | 3.33 | 1.82 | 2.46 | 2.12 | 1.07 | 2.69 | 18.39 | 16.01 | 0.17 |
| Total | 98.18 | 98.82 | 99.59 | 96.48 | 99.30 | 96.77 [2] | 99.33 [3] | 100.32 [4] | 99.10 [5] |
| **Structural Formula (apfu) Calculated on the Basis of O = 8 Atoms [6]** | | | | | | | | | |
| Li | 0.00 | 0.00 | 0.00 | 0.00 | 0.00 | 0.00 | 0.04 | 0.01 | 0.00 |
| Mn | 0.84 | 0.86 | 0.88 | 1.02 | 0.98 | 0.81 | 0.68 | 1.04 | 0.88 |
| $Fe^{2+}$ | 0.17 | 0.14 | 0.12 | 0.00 | 0.02 | 0.19 | 0.28 | 0.11 | 0.12 |
| $\Sigma A$ | 1.01 | 1.00 | 1.00 | 1.02 | 1.00 | 1.00 | 1.00 | 1.16 | 1.00 |
| $Fe^{3+}$ | 0.14 | 0.16 | 0.15 | 0.13 | 0.03 | 0.06 | 0.43 | 0.30 | 0.10 |
| Ti | 0.08 | 0.02 | 0.11 | 0.08 | 0.12 | 0.08 | 0.12 | 0.06 | 0.14 |
| Sn | 0.46 | 0.47 | 0.48 | 0.39 | 0.74 | 0.57 | 0.17 | 0.40 | 0.56 |
| Ta | 0.29 | 0.34 | 0.26 | 0.36 | 0.09 | 0.17 | 0.26 | 0.28 | 0.20 |
| $\Sigma B$ | 0.97 | 0.99 | 1.00 | 0.96 | 0.98 | 0.88 | 0.98 | 1.04 | 1.00 |
| Nb | 0.64 | 0.63 | 0.56 | 0.66 | 0.24 | 0.25 | 0.78 | 0.50 | 0.39 |
| Ta | 1.27 | 1.32 | 1.39 | 1.28 | 1.76 | 1.67 | 0.76 | 1.06 | 1.61 |
| W | 0.09 | 0.05 | 0.05 | 0.06 | 0.03 | 0.08 | 0.46 | 0.44 | 0.00 |
| $\Sigma C$ | 2.00 | 2.00 | 2.00 | 2.00 | 2.03 | 2.00 | 2.00 | 2.00 | 2.00 |

The representative electron probe microanalysis (WK-208 and WK-222) and the average composition of 17 samples (WK-av) of wodginite from the Kester deposit are presented. Average composition of tungsten-bearing wodginite from other deposits is shown: 1—Dajishan, China (2 samples) [12]; 2—Keivy, Kola Peninsula [3]; 3—Pendalras, India (12 samples) [20]; 4—Songshugang, China (3 samples) [11]. Wwdg—"wolframowodginite" (average of 5 samples) [19]; Wdg—wodginite (average of 206 samples, according to Alfonso et al. (2018), Gaafar (2014), Kesraoui and Nedjari (2002), Huang et al. (2002), Moussa et al. (2021), Wu et al. (2017), Zhu et al. (2015) [5,6,10–13,15], and author data). Asterisk "*" denotes wodginite in Li-F granites. [1] $FeO_t$ = FeO + $Fe_2O_3$. [2] Including CaO 0.34, $ZrO_2$ 1.07, and $HfO_2$ 0.27. [3,4] Including $Sc_2O_3$ 0.79 and 0.05. [5] Including $Sc_2O_3$ 0.02, $ZrO_2$ 0.02, and $HfO_2$ 0.03. [6] Values are calculated by means of stoichiometry; the cations of $Fe^{2+}$ and $Fe^{3+}$ were calculated according to the methods of Ercit et al. (1992) [24].

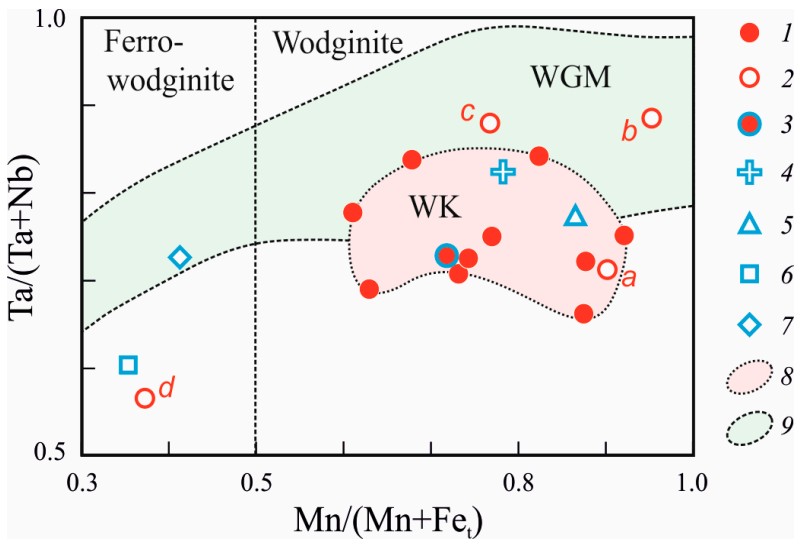

**Figure 3.** Chemical compositions of tungsten-bearing wodginite in Li-F granites from the Kester

deposit and from other deposits and wodginite-group minerals in the world plotted in the columbite quadrilateral (atomic proportions): 1—Tungsten-bearing wodginite from the Kester deposit. 2—Tungsten-bearing wodginite from other deposits: *a*—Dajishan, China (average of 2 samples) [12]; *b*—Keivy, Kola Peninsula [3]; *c*—Pendalras, India (average of 12 samples) [20]; *d*—Songshugang, China (average of 3 samples) [11]. 3—"Wolframowodginite" (average of 5 samples) [19]. 4—Wodginite (70 samples) [5,6,10–13,15]. 5—Titanowodginite (7 samples) [5,6,29]. 6—Ferrowodginite (6 samples) [8,11]. 7—Ferrotitanowodginite (2 samples) [14]. 8—The fields of wodginite from the Kester deposit (WK). 9—The fields of wodginite-group minerals (WGMs) [30]. Compositions are shown in Table 1.

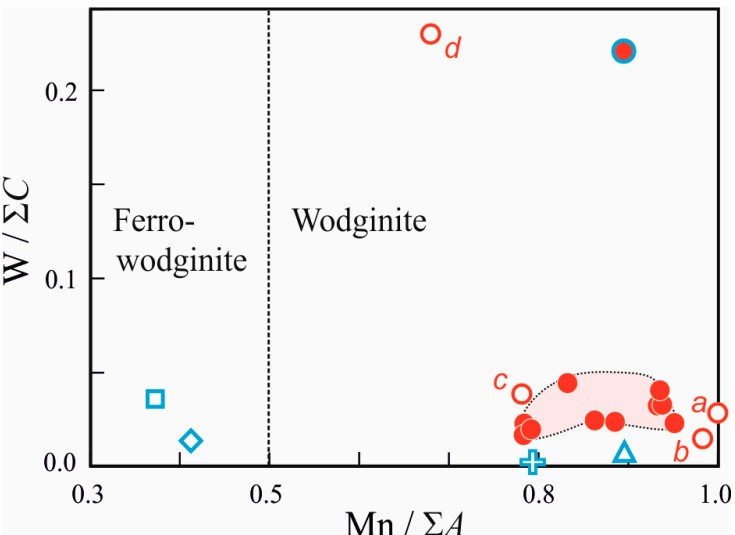

**Figure 4.** Compositions of tungsten-bearing wodginite in Li-F granites from the Kester deposit and from other deposits and wodginite-group minerals in the world in the quadrilateral of wodginite classification Mn/$\Sigma A$ versus W/$\Sigma C$ (atomic proportions). $\Sigma A$, $\Sigma C$—the sums of cations in positions $A$ ($Mn^{2+}$, $Fe^{2+}$, Li, Ca) and $C$ ($^B$Ta, Nb, W). For symbols, see Figure 3.

## 5. Discussion

### 5.1. Tungsten in Accessory Minerals of Lithium-Fluoric Granites from the Russian Far East

In the 20th century, wodginite was known as a mineral typical of pegmatite. Since 2002, it has been found in rare-metal LFGs [4–6,8,11,12,15,16]. The reasons for the late discovery of wodginite in granites are its insignificant size (less than 100 μm) and resemblance to tantalite. In Russian granites, wodginite was detected only in the Voznesenskoye deposit (Primorye) [29].

Under specific conditions, tungsten can act as an element typical of residual pegmatite melt (solution), which leads to the formation of tungsten-bearing wodginite in rare-metal granites [3]. When the crystallization of rare-metal granite melt proceeds under subsolidus conditions and is followed by enrichment with tungsten, two trends run in parallel: (1) manganocolumbite → manganotantalite → wodginite + titanowodginite → cassiterite; (2) W-columbite-tantalite → W-bearing ixiolite → wolframite [6]. The above-indicated resemblance among wodginite from the Kester deposit, wolframowodginite, and titanowodginite proves that in specific minerogenetic processes, these trends can merge into one to form Ti- and W-bearing wodginite: tungsten-bearing columbite + tungsten-bearing tantalite → W-bearing ixiolite + titanium-bearing wodginite + tungsten-bearing wodginite → cassiterite + wolframite.

The discovery of tungsten-bearing wodginite in LFGs from the Kester deposit widens the list of accessory minerals in granites that are enriched with tungsten (wolframite, W-bearing ixiolite, columbite, tantalite-(Mn), and microlite) [27]. Furthermore, there is a confirmation of the previously mentioned regularity; the tungsten-related geochemical specialization of LFGs from the Russian Far East is expressed in a wide collection of tungsten

and tungsten-bearing accessory minerals [27,31]. Wodginite from the Voznesenskoye deposit in the Far East also contains about 1.9% $WO_3$ [29].

### 5.2. Isomorphism of Wodginite and the Possibility of Identifying "Wolframowodginite"

Minerals of wodginite groups (WGMs), $ABC_2O_8$ (Z = 4), stand out among tantalo-niobates for their maximally ordered layered structure, *ABA–CCC–BAB*, attributed to the presence of $^B\text{Sn}^{4+}$-cations. Wodginite can be considered as a maximally ordered ixiolite with a quadrupled elementary cell [24,32]. The composition of the components in WGMs is as follows: $A$ = ($Mn^{2+}$, $Fe^{2+}$, Li, Ca, □), $B$ = ($Sn^{4+}$, Ti, $Fe^{3+}$, Ta, Sc, Zr), and $C$ = (Ta, Nb) [3,19,24]. Polyelement isomorphism determines the distinguishing of different mineral species: wodginite, $MnSnTa_2O_8$; titanowodginite, $MnTiTa_2O_8$; ferrowodginite, $FeSnTa_2O_8$; ferrotitanowodginite, $FeTiTa_2O_8$; (5) lithiowodginite, $LiTaTa_2O_8$; (6) tanta-lowodginite, $(Mn_{0.5}\square_{0.5})TaTa_2O_8$.

In pegmatite from Separated Rapids (Canada), one more hypothetical mineral species, called "wolframowodginite", was found. However, due to insufficiency of available data, the mineral species was not proven [19]. "Wolframowodginite" was also found in LFGs from the Songshugang deposit in China [11]. The concentration of $WO_3$ in "wolframowodginite" in granites is, on average, 18.39% (maximum of 23.93%) [11], and it is 16.01% (up to 34.63%) in pegmatites [19]. "Wolframowodginite" is enriched with lithium, the content of which is second only to lithiowodginite: 0.21% (maximum of 0.29%) $Li_2O$ in granites and 0.06% (maximum of 0.16%) in pegmatites. The finding of wodginite with a high content of $WO_3$ in granites from China and Spain [12,13] and in pegmatites from Russia and India [3,20] confirms the possibility of describing a new mineral called "wolframowodginite" (Table 1).

The similarity in structure between wodginite and wolframite was experimentally detected [33]. The presence of structural elements typical of wolframite and W-bearing ixiolite in the structure of wodginite was previously noted by Ercit et al. [24]. It was concluded that the same role of Sn and W in the layered structure of ixiolite is the basis of the wodginite structure [3]. Wodginite contains tungsten in position C, dominated by Ta and Nb that form octahedral sheets of $NbO_6$ and $TaO_6$—the most stable element in the structure of WGMs. The prevalence of Ta–O octahedra is the key for the highly ordered structure of wodginite [3]. Isomorphism between Ta and Nb is limited by the value of 8 cation apfu [24]. A correlation between $WO_3$ and (FeO + $Fe_2O_3$) was previously revealed by Sarbajna et al. (2010) [20], Wu et al. (2017) [12], and Alfonso et al. (2018) [13]. For ferruginous varieties of WGMs, the following mechanism of tungsten implantation was suggested: $^B[\text{Sn}^{4+}] + {}^C[\text{Ta}^{5+}] \leftrightarrow {}^B[\text{Fe}^{3+}] + {}^C[\text{W}^{6+}]$; for manganiferous varieties, the mechanism is $^B[\text{Sn}^{4+}] + 2^C[\text{Ta}^{5+}] \leftrightarrow {}^B[\text{Mn}^{2+}] + 2^C[\text{W}^{6+}]$; for Li-bearing wodginite, the mechanism is $2^A[\text{Li}^+] + 2^A[\text{Mn}^{2+}] \leftrightarrow {}^C[\text{W}^{6+}]$ [19] (Figures 3 and 4; Table 1).

In wodginite from the Kester deposit, the $WO_3$ content is 1.23%–3.33%. This allows us to put it on the list of tungsten-bearing species of wodginite. Wodginite is characterized by tin deficiency, which is not an obstacle to mineral crystallization and can be compensated by impurities of Ti, W, and $Fe^{3+}$. The dots of "wolframowodginite" and tungsten-bearing wodginite from the Kester deposit are partially located within the field of WGMs and expand its area (Figure 3). Like previously mentioned findings of tungsten-bearing wodginite, the described mineral from Yakutia confirms the possibility of identifying "wolframowodginite" (Table 1).

### 5.3. "Wolframowodginite" Indicating the Late Stage of Magmatic Crystallization of Pegmatites and Granites

WGMs serve as indicators of rare-metal petrogenesis. They are considered to be markers of Li-F granites and pegmatites that have crystallized in the late magmatic stage [3,6,29]. Tungsten with tin, being typical elements of residual rare-metal melts, isomorphically substitute niobium in late-magmatic wodginite (Table 1) [3,11,19]. Tungsten-bearing wodginite emerges from magma enriched with F, Mn, Ta, Sn, and W as a result of ultimate magmatic

fractionation to the extreme values of Ta/(Ta + Nb) and Mn/(Mn + Fe) [4,11,15,19,34] (Figures 3 and 4). It was experimentally established that wodginite crystallizes either from granitic melt at temperatures of 700–800 °C [34] or from salt (hydrosilicate) melt enriched with Ta and Sn [12].

The evolution of rare-metal granitic magma determines the formation of the standard sequence of tantalo-niobates: columbite-(Fe) → columbite-(Mn) → tantalite-(Mn) → wodginite → microlite → cassiterite [3,11,19,30]. In granites from the Kester deposit, this sequence can be observed in its complete form. The apical parts of LFG intrusions demonstrate the concentration of minerals rich in Mn-Ta-Sn-W [23]. Wodginite is a part of ores containing Ta, Sn, Li, and W from the Kester and Voznesenksoye deposits [29].

The literature describes two main parageneses of wodginite: (1) microinclusions in cassiterite, which are the results of the solid solution of Ta-bearing cassiterite being decomposed; (2) association with columbite, tantalite, microlite, and cassiterite—the result of direct crystallization from rare-metal melts. Both parageneses were found at the Kester deposit (Figure 2).

Tungsten-containing wodginite and tantalite-(Mn) are contemporaneous, considering the mixture between two Nb–Ta oxides (Figure 2).

Under subsolidus conditions, the crystallization of rare-metal granitic melt enriched with tungsten develops in two directions: (1) columbite-(Mn) → tantalite-(Mn) → wodginite + titanowodginite → cassiterite; (2) W-bearing columbite-tantalite → W-bearing ixiolite → wolframite [6]. In the case of the Kester deposit, these two trends merge into one: W-bearing columbite-(Fe) + W-bearing tantalite-(Mn) → W-bearing ixiolite + W-bearing wodginite → cassiterite + wolframite. In this way, tungsten-bearing wodginite from Yakutia is an example of rare heterovalent isomorphism and an indicator of rare-metal granites and pegmatites bearing tin and tantalum. The possibility of such isomorphism is determined by the complex composition of mineral-forming rare-metal magma and the properties of the wodginite structure, which admits the coexistence of elements with variable valence (Mn, Fe, Sn, W, and Ti).

## 6. Conclusions

The identification of tungsten-bearing wodginite at the Kester deposit confirms the previously established regularity, namely, the tungsten geochemical specialization of Li-F granites from the Russian Far East, characterized by the widespread presence of tungsten and tungsten-bearing accessory minerals.

Finding wodginite with a high content of $WO_3$ in Yakutia, in conjunction with findings of similar minerals in granites from China and Spain and in pegmatites from Canada, Russia, and India, supports the possibility of identifying a new mineral called "wolframowodginite".

Tungsten-bearing wodginite is an indicator of tin and tantalum-bearing rare-metal granites and pegmatites.

**Author Contributions:** Conceptualization, V.I.A.; methodology, V.I.A.; software, V.I.A.; validation, V.I.A.; formal analysis, V.I.A.; investigation, V.I.A.; resources, V.I.A.; data curation, V.I.A. and I.V.A.; writing—original draft preparation, V.I.A.; writing—review and editing, I.V.A.; visualization, V.I.A. and I.V.A. All authors have read and agreed to the published version of the manuscript.

**Funding:** The study was funded by Russian Foundation for Basic Research (project No. 20-15-50064).

**Institutional Review Board Statement:** Not applicable.

**Informed Consent Statement:** Not applicable.

**Data Availability Statement:** Data is unavailable due to privacy restrictions.

**Acknowledgments:** The authors extend their appreciation to Saint-Petersburg Mining University for analytical studies.

**Conflicts of Interest:** The authors declare no conflict of interest.

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
