# Peer review of "Tungsten-Bearing Wodginite from the Kester Deposit, Eastern Siberia, Russia"

_minerals, doi:10.3390/min13020231_

Round 1

Reviewer 1 Report

The proposed paper concerns a peculiar association with rare metal (Ta-Nb in the first place, associated with W, Sn, Mn and Ti) ores in Li-F granites from Eastern Siberia. In particular, the study focused on the mineral chemistry of wodginite minerals, a group of Ta phases whose W contents were poorly documented in literature. Although designedly limited in scope (W contents in wodginite) and methods (SEM-EDS and EPM analyses only), the study provides interesting new data for further studies on this group of minerals and their petrogenetic role, and it offers new exploration tools for tungsten and rare metals granite-related ores in Russian Far East and similar areas worldwide. I found the paper well-organized, the methods appropriate to the scope and the data generally well-presented (but I made some observations, particularly on figures and captions, see the attached file); the discussion may appear at times speculative (i.e., the “wolframowodginite” paragraph, in absence of more strict crystallographic data on studied samples), but is always on focus. The text is generally well-written, but sometimes with some obscure expressions, thus I recommend a further revision for English.

Reviewer 2 Report

Line 134 135:  day surface (you mean outcrop I presume) /wells (hole)? Are you sure about English?

Line 178 : totaNowodginite

Line 2013 raRe- metal granite

 Figure 3 and 4: if possible (without compromising easy reading) to show the real data, of each sample analyses, and not only the calculated mean.

Reviewer 3 Report

The subject of the article concerns a very important aspect of the development of the global economy, which is the possibility of obtaining critical metals. The article is focused on determining the occurrence of tungsten compounds in a specific deposit. The scientific aspect, however, is not very clear. Demonstrating the presence of tungsten in this deposit, however, does not give any information as to the possibility of recovering this metal from this raw material.

I don't care to edit the article. correctly.  The number of self-citations is acceptable.

The conclusions are clear, but I would like to extend them based on a discussion of the results.

Please explain the index "*" in the description of the components in table 1.
